# The Recovery of Bioactive Compounds from Olive Pomace Using Green Extraction Processes

Marina Stramarkou [1,*], Theodora-Venetia Missirli [1], Konstantina Kyriakopoulou [1], Sofia Papadaki [1], Athanasios Angelis-Dimakis [2] and Magdalini Krokida [1]

[1] Laboratory of Process Analysis and Design, School of Chemical Engineering, National Technical University of Athens, 9 Iroon Polytechneiou St. Zografou Campus, 15780 Athens, Greece; mkrok@chemeng.ntua.gr (M.K.)

[2] Department of Chemical Sciences, School of Applied Sciences, University of Huddersfield, Queensgate, Huddersfield HD1 3DH, UK

[*] Correspondence: m_stramarkou@hotmail.com

**Abstract:** In this study, solid olive mill waste (SOMW) was used to obtain antioxidant compounds using solid–liquid extraction. The effect of different extraction methods, namely microwave-assisted extraction (MAE), ultrasound-assisted extraction (UAE), Soxhlet, and conventional solvent extraction, on the yield, total phenolics, and total antioxidant activity of SOMW extracts was investigated. Untreated and dried SOMW were subjected to extraction with water and methanol. The antioxidant activity of the extracts was evaluated using the DPPH assay, while their total phenolic content was measured using the Folin–Ciocalteu method. For the characterisation of the extracts, HPLC-DAD analysis was performed. The results showed that the extraction yield was significantly influenced ($p < 0.05$) by the solvent used, the material treatment prior to extraction, the moisture content of SOMW samples, and the extraction time. The optimised parameters were water, as the extraction solvent, and MAE as the extraction technique (extraction temperature of 50 °C and time of 1 h). The evaluation of the antioxidant activity of the extracts indicated that phenolics were the dominant bioactive compounds. The extracts were found to be rich in several hydroxytyrosol derivatives. Therefore, SOMW can be a valuable resource for bioactive compounds using conventional and innovative extraction techniques.

**Keywords:** accelerated solar drying; air drying; antioxidant potential; HPLC-DAD; microwave-assisted extraction; olive mill pomace; polyphenols; ultrasound-assisted extraction



## 1. Introduction

Olive oil production is one of the most important agricultural industries of the Mediterranean region, covering 97% of worldwide olive oil production and employing 3.3 million olive growers, which is equal to one-third of the total EU farmers [1,2]. According to the latest data from FAOSTAT, more than 3.37 million tons of olive oil annually were produced in 2020, which were primarily intended for human consumption, showing the importance of the sector in the economy and the social activity of the producing countries [3].

However, during the olive oil extraction process, various by-products, such as wastewater and solid residues, are generated in vast quantities, causing severe environmental problems [4]. During the olive milling season, the olive mill production generates not only liquid but also solid (olive pomace) mill waste [5]. Over the years, solid olive mill waste (SOMW) has been considered a major environmental polluting factor that requires effective handling, due to its high organic load, low pH, and elevated salt and phenolic content [6,7]. Thus, the effective management of SOMW is necessary, especially for the Mediterranean countries that have to cope with seasonal operations and the low-scale capacity of olive mills, making its treatment inefficient and costly [8,9]. Greece is one of four European countries that produce 12.5 million tonnes of olive oil in total, accounting for more than

95% of the total EU production [10]. The majority of olive mills operating in Greece are three-phase centrifugal systems. A few old-style presses are also preserved. Two-phase oil mills are not very widespread in Greece. Our search aims to develop strategies for the valorisation of SOMW, using the value chain of Greek olive oil production as a case study. Therefore, this study focused on the valorisation of pomace from three-phase olive mill systems [11]. Among the current solutions, composting [12] and anaerobic digestion are commonly used [13]. However, these treatments do not properly exploit the valuable phytochemical content and phenolic compounds of SOMW.

Phenolic compounds, especially hydroxytyrosol, tyrosol, and seicoroids, are some of the most valuable fractions of SOMW possessing strong antioxidant, antimicrobial, and phytotoxic properties and having potential industrial applications, such as antioxidants, fertilisers, and antibacterial drugs, as well as gelling and stabilising agents in food products [9,14]. Phenolic compounds are abundant in SOMW since, due to their hydrophilic nature, only 2% are transferred to olive oil during olive oil extraction, whereas 98% remain in the SOMW [5]. Therefore, the valorisation of the SOMW through the optimisation of the recovery of high-value-added compounds and their commercial use is of great importance [14].

Conventional and Soxhlet extraction methods represent the common lab-scale extraction methods for the recovery of phenolic compounds. Conventional extraction is a widely used method for obtaining bioactive compounds from plants and other natural sources. It involves the use of solvents, such as ethanol or methanol, to extract the desired components from the plant material. This method is relatively simple and cost-effective, making it a popular choice in many research studies. On the other hand, Soxhlet extraction is a more specialised extraction technique that allows for the efficient extraction of target compounds from solid materials. It utilises a continuous extraction process, where a solvent is repeatedly circulated through a sample. This method is particularly suitable for extracting compounds that have low solubility in the chosen solvent or for samples with complex matrices. Soxhlet extraction is known for its ability to yield high extraction efficiencies, making it advantageous when dealing with limited sample quantities. In summary, both conventional extraction and Soxhlet extraction methods have been widely used for the extraction of antioxidants with antiradical effects. These methods enable researchers to obtain natural extracts that possess significant antioxidant activity, providing valuable insights for the development of antioxidant-based therapies or dietary supplements to combat oxidative-stress-related diseases [15,16].

However, several disadvantages, such as the loss of compounds due to hydrolysis and oxidation, the high requirement of time and the use of large volumes of organic solvents, limit their use and promote more efficient and environmentally friendly techniques for a rapid analytical-scale extraction process, such as ultrasound-assisted extraction (UAE) and microwave-assisted extraction (MAE) [17–19]. The release of bioactive compounds from the plant tissue is improved, and the solvent's access to the biomass is facilitated thanks to the phenomena of cavitation during UAE and selective microwave dielectric heating during MAE [19]. By comparing the antiradical effects of the extracts obtained through different extraction methods, researchers can gain insights into the efficiency of each technique in preserving and extracting antioxidant compounds. This information is crucial for selecting the most appropriate extraction method when targeting specific antioxidant-rich plant sources.

The objective of the present work is to study the effect of different extraction methods on the recovery of compounds with high antiradical activity from untreated and dried SOMW. For the drying treatment of SOMW, accelerated solar-drying (ASD) and air-drying (AD) methods were selected due to their application in the agri-food industry and their low investment and operating cost [20]. Untreated and dried SOMW samples were extracted using traditional and green extraction techniques, and extraction yield, antiradical activity, total phenolic content, extraction kinetics, and chemical composition were determined.

## 2. Materials and Methods

### 2.1. Plant Material

SOMW was supplied by a continuous three-phase olive processing plant located in Messinia (South Greece, Peloponnese) and was obtained after olive oil production in the first months of 2022 from olives of the cultivated tree variety Koroneiki I-38. Oil extraction and pomace sample collection were performed on the same day. The sample contained $66.10 \pm 1.31\%$ moisture content and was immediately stored at $-30\,°C$ in order to avoid the enzymatic degradation of the polyphenols until further use.

### 2.2. Chemicals and Reagents

All the reagents and solvents used in the extractions were of analytical grade. Gallic acid (GAE), Folin–Ciocalteu reagent, and 2,2-diphenyl-picrylhydrazyl (DPPH) were purchased from Sigma-Aldrich (Gillingham, UK). Water, methanol, and sodium carbonate ($Na_2CO_3$) were purchased from Fisher Scientific (Leicestershire, UK). For HPLC analysis, methanol (HPLC grade), acetic acid and standard solutions of hydroxytyrosol, were purchased from Sigma-Aldrich Company Ltd. (Gillingham, UK).

### 2.3. Drying Experiments

Untreated samples of SOMW were dried using two different drying methods: air drying (AD) and accelerated solar drying (ASD).

Regarding the first method, the samples were placed on perforated trays, perpendicular to the airflow in an experimental air dryer, consisting of airflow rate control, heating control, humidity control, and drying test compartments. The experiments were conducted at three temperatures (35, 50, and 70 ($\pm 1$) $°C$) and 1.0 m/s air velocity for 24 h, under atmospheric pressure.

In the case of ASD, the samples were placed in a laboratory-scale accelerated artificial solar dryer with a total light source of $1000\ W \times m^{-2}$ (8 OSRAM Ultra-Vitalux lamps, 300 W) at a temperature of $50 \pm 2\,°C$. The total exposure time was 24 h.

All the drying experiments were conducted in triplicate, and after drying, the dried products were stored at $-30\,°C$ in plastic containers until being subjected to extraction.

Moisture Content Determination

The moisture content of the untreated and dried SOMW was determined according to AOAC (1980) [21] using a vacuum oven (Sanyo Gallenkamp PLC, Leicester, UK) maintained at $70 \pm 0.2\,°C$, until a constant weight was achieved. The moisture content calculation was based on the following equation:

$$MC_{(wet\ basis)} = \frac{(m_w - m_d)}{m_w} \tag{1}$$

where $MC$ is the moisture content on a wet basis (g/g), $m_w$ is the wet weight (g), and $m_d$ is the dried weight of the sample (g). The moisture content of the SOMW samples was measured at predetermined intervals during drying. Experiments were conducted in triplicate.

### 2.4. Extraction Process

Untreated and dried SOMW samples were extracted through microwave-assisted extraction (MAE), ultrasound-assisted extraction (UAE), conventional extraction (CE), and Soxhlet extraction (SE) using the polar solvents of water and methanol in order to recover bioactive compounds with high antiradical activity. Prior to extraction, the dried samples were ground to increase the active surface area. The mean dimension of the ground pomace particles was 900 μm. All the extraction experiments were conducted in triplicate, and their setup conditions are presented in Table 1.

**Table 1.** Description of extraction experiments.

| | | Extraction Experiments | | |
|---|---|---|---|---|
| **Solvents** | **MAE** | **UAE** | **CE** | **SE** |
| Water | 1 g/120 mL, 200 W, 50 °C, 5, 10, 20 and 60 min | 0.5 g/50 mL, 25 kHz, 450 W, 25 °C, 5, 10, 20 and 60 min | 0.5 g/100 mL, 25 °C, 24 h | 2 g/100 mL, 3–4 h |
| Methanol | 1 g/ 120 mL, 200 W, 50 °C, 5, 10, 20 and 60 min | 0.5 g/50 mL, 25 kHz, 450 W, 25 °C, 5, 10, 20 and 60 min | 0.5 g/100 mL, 25 °C, 24 h | 2 g/100 mL, 5–6 h |

### 2.4.1. Microwave-Assisted Extraction (MAE)

MAE was carried out in an XO-SM50 Ultrasonic Microwave Reaction System (Nanjing Xianou Instruments Manufacture Co., Ltd., Nanjing, China). Briefly, 1 g of untreated or dried SOMW samples were extracted with 120 mL of water or methanol using the fluctuating radiation of 200 W to keep the temperature steady at 50 °C. Since extraction time is a key parameter of the extraction yield, four different extraction times were examined: 5, 10, 20, and 60 min.

### 2.4.2. Ultrasound-Assisted Extraction (UAE)

UAE was carried out in an XO-SM50 Ultrasonic Microwave Reaction System (Nanjing Xianou Instruments Manufacture Co., Ltd., Nanjing, China). Briefly, 0.5 g of untreated and dried SOMW samples were placed in a beaker with 50 mL of water or methanol, operating at 25 kHz frequency, 450 W power, and 25 °C temperature. Samples were collected at 5, 10, 20, and 60 min of extraction.

### 2.4.3. Conventional Extraction (CE)

In the conventional extraction, 0.5 g of untreated and dried SOMW samples were placed in the extraction beaker with 100 mL of the desired solvent (water and methanol). The samples were subjected to agitation using a magnetic stirrer for a total duration of 24 h at 25 °C.

### 2.4.4. Soxhlet Extraction (SE)

Briefly, 2 g of untreated and dried SOMW samples were placed into an extraction thimble with 100 mL of water and methanol in the extraction flask in a lab-scale extraction apparatus. Water was refluxed at 3–4 h and methanol at 5–6 h till the completion of five to seven extraction circles.

The CE and SE methods were chosen in order to compare the yielding and selectivity of MAE and UAE with conventional applied techniques.

### 2.4.5. Extraction Yield (*EY*)

The *EY* is expressed as a percentage of the weight of the obtained dried extract relative to the initial dried matter of the sample used for the extraction, as described in the following equation:

$$EY\,\% = \frac{mass\,of\,dry\,extract}{mass\,of\,dry\,matter} \times 100 \qquad (2)$$

### 2.5. Antiradical Scavenging

A thin-layer chromatographic (TLC) method was applied for the preliminary screening of free radical scavengers of the extracts, as proposed in a previous study [22]. The extracts that exhibited a rapid change in colour (white spots on a purple background) were considered active. Aqueous and methanolic extracts recovered at longer treatment times exhibited instant discoloration; therefore, they were chosen for further investigation.

The antiradical capacity of the selected SOMW extracts was determined using the stable radical 2,2-diphenyl-1-picrylhydrazyl (DPPH). The analytical procedure was performed as proposed by Kyriakopoulou et al. (2013) using a UV–Vis spectrophotometer

(Spectrometer UV-M51, Bel Photonics, Sao Paulo, Brazil) to measure the absorbance at 515 nm until 30 min of reaction [22]. The changes in the absorbance of appropriately diluted SOMW samples were measured at 25 °C, while the DPPH concentration in the reaction medium was determined with the use of a calibration curve at 515 nm. The percentage of the remaining DPPH (%DPPH rem) was calculated using Equation (3) as follows:

$$\text{DPPH rem \%} = \frac{[\text{DPPH}]_t}{[\text{DPPH}]_{t0}} \times 100 \tag{3}$$

All measurements were performed in duplicate.

### 2.6. Total Phenolic Content

The concentration of the total phenolics of SOMW extracts was determined using a UV–Vis spectrophotometer (Spectrometer UV-M51, Bel Photonics, Sao Paulo, Brazil) and the Folin–Ciocalteu method [20]. The measurements were conducted in duplicate, and the total phenolic content is expressed as ppm gallic acid equivalent (GAE) in dry extract.

### 2.7. Extraction Kinetics of Total Polyphenols

For the determination of extraction kinetics, the phenolic content of the samples was regularly measured during the treatment. The extraction curves (concentration of total polyphenols vs. time) were validated using the Peleg (1988) sorption model, which, in the case of solid–liquid extraction processes, assumes the following form:

$$C_{(t)} = C_0 + \frac{t}{K_1 + K_2 \cdot t} \tag{4}$$

where $C(t)$ is the concentration of the total phenolic compounds expressed in mg of gallic acid equivalents/g dry biomass at time $t$ ($mg_{GAE}/g_{db}$); $t$ is the extraction time (min); $C_0$ is the initial concentration of total phenolic compounds at time $t = 0$ ($mg_{GAE}/g_{db}$); $K_1$ is Peleg's rate constant ($min \cdot g_{db}/mg_{GAE}$) and expresses the rate of the TPC extraction; and $K_2$ is Peleg's capacity constant ($g_{db}/mg_{GAE}$) and is related to maximum attainable TPC content. Since $C_0$ in all experimental runs was zero, Equation (4) is modified to the following form:

$$C_{(t)} = \frac{t}{K_1 + K_2 \cdot t} \tag{5}$$

The extraction rate ($B_0$) at the very beginning ($t = t_0$) expressed in mg of gallic acid equivalents/g dry biomass·min ($mg_{GAE}/g_{db} \cdot min$) relates to the Peleg rate constant $K_1$ as follows:

$$B_0 = \frac{1}{K_1} \tag{6}$$

where the capacity constant $K_2$ relates to the maximum of *EY*, i.e., the equilibrium concentration of the total extracted polyphenols (*Ce*) expressed in mg of gallic acid equivalents/g dry biomass ($mg_{GAE}/g_{db}$), and when $t \rightarrow \infty$, Equation (7) gives the relations between equilibrium concentration and $K_2$ constant as follows:

$$C|_{t \rightarrow \infty} = C_e = \frac{1}{K_2} \tag{7}$$

The parameters of the modified Peleg's model were determined from experimental data using nonlinear regression. The accordance between experimental data and the calculated value was established using the correlation coefficient (*q*) and the root-mean-squared deviation (*RMSD*).

*2.8. Identification of Phenolic Compounds Using High-Performance Liquid Chromatography (HPLC)*

The HPLC analysis of the selected extracts was conducted on a Shimadzu liquid chromatograph system (Shimadzu Corp, Kyoto, Japan) equipped with a quaternary pump, a vacuum degasser, an autosampler, a diode array detector (DAD), a tunable UV–Vis detector, and a Luna C18 column (250 mm × 4.6 mm, 5 μm) (Phenomenex, Australia). The results were acquired and processed using the Shimadzu Workstation CLASS-VP 6.12 software (Shimadzu Corp). A gradient solvent system was employed as proposed by Jerman et al. (2010), with solvent A being water–acetic acid (95:5, *v*/*v*) and solvent B being methanol. The elution profile (*v*/*v*) of solvent B was the following: 0 min, 5%; 3 min, 10%; 18 min, 25%; 19 min, 29%; 24 min, 30%; 30 min, 31%; 31 min, 35%; 41 min, 45%; 51 min, 55%; 61 min, 65%; 67 min, 100%; and 70–80 min, 5%. The column was held at 25 °C, and the flow rate was 1.0 mL/min. All the prepared solutions were filtered through 0.45 mm PTFE filters (MachereyNagel, Düren, Germany). A sample volume (20 μL) was injected and DAD signals were recorded at 280, 320, and 365 nm [23]. The chromatographic identification and confirmation of phenolic compounds were based on comparing the retention times with the absorption spectrum data from the literature [23–27].

Prior to HPLC, the extracts were dried using a freeze-dryer (Leybold–Heraeus GT 2A, Koln, Germany) to avoid any further thermal degradation, and the dried extracts were redissolved in 1 mL of acidic HPLC eluent ($H_2O$/ $CH_3COOH$, 95:5, *v*/*v*). The diluted samples were filtered through 0.45 μm PTFE filters and analysed using HPLC.

*2.9. Statistical Analysis*

A two-way analysis of variance (ANOVA) with a post hoc Tukey HSD (honestly significant difference) test was applied to detect the significant differences in the calculated *EY* dependent on the drying treatment and extraction time, while a one-way ANOVA was used to determine the significant differences among the different dried samples in terms of antiradical activity and total phenolic content. Analyses were performed with the STATISTICA software (Version 13.6 StatSoft® Inc., Palo Alto, CA, USA). A significance level of *a* = 0.05 was selected, and differences were considered to be significant at *p* < 0.05.

## 3. Results

*3.1. Moisture Content*

Drying is considered one of the most commonly used methods for food preservation [20]. Except for diminishing enzymatic reactions or reducing microbiological spoilage, drying also reduces the weight and volume of products, facilitating their transportation and storage [28]. Therefore, drying is considered an essential step prior to any valorisation technique in order to avoid any modification by microorganisms in SOMW due to its highwater content.

In the present study, drying under different temperatures led to variations in the final moisture content of olive oil's solid by-product. The average moisture content of the samples after the drying experiments was estimated up to 3.58 ± 0.28% for ASD, 6.90 ± 1.05% for AD 35 °C, 2.78 ± 0.03% for AD 50 °C, and 1.30 ± 0.12% for AD 70 °C, where ± shows the standard deviation between the replicates. Apart from the reduction in the moisture content of the material, drying causes several physical, structural, and chemical changes, especially under intensive conditions, such as elevated temperature. Phenolic compounds, which represent compounds with high biological value, can be lost or deteriorated at high-temperature treatment [20,29]. Therefore, dried and untreated SOMW samples were subjected to several processes of extraction following the drying process in order to evaluate how the drying treatment can affect their total phenolic content and their antiradical activity.

*3.2. Extraction Yield (EY)*

In general, the *yield* of extraction processes is directly influenced by several parameters, such as the type of solvent, the method of extraction, the processing time and temperature, and the moisture content of the sample.

Regarding the first parameter, the polarity of the selected solvent affected the solubility of the compounds of the sample and finally the *EY*. Methanol is a commonly used solvent for the recovery of bioactive compounds from olive mill by-products [25]. On the other hand, water constitutes a preferable solvent due to its nontoxic and eco-friendly character, its inexpensive features, and its high extractability potential [30]. Other advantages of the use of water as a solvent include the limitation of the use of organic solvents and the absence of any restriction for human consumption [31–33].

Furthermore, regarding the method of extraction, in recent years, conventional extraction methods, such as CE and SE, are replaced by novel techniques such as MAE and UAE. The main assets of the latter techniques are their reduced solvent volume and extraction time, which contribute to the elimination of the degradation and oxidation of phenolic compounds and the improvement of *EY* [19,34].

The results of the *EY*s of each extraction technique, namely MAE, UAE, CE, and SE, for both untreated and dried SOMW samples, are presented in the following paragraphs.

3.2.1. Microwave-Assisted Extraction (MAE)

The results of the *EY* for MAE using water and methanol are presented in Figure 1a,b, respectively.

As observed, the drying pretreatment of the samples, the used solvent, and the extraction time resulted in great variations in the *EY* with significant differences ($p < 0.05$).

Specifically, as shown in Figure 1a, the untreated SOMW samples presented the highest *EY*, up to $14.37 \pm 0.36\%$ after 60 min of extraction, followed by the samples dried with AD at 35 °C ($6.38 \pm 0.32\%$). The higher *EY* of the untreated and dried samples with mild drying techniques (ASD and AD 35 °C) revealed that MAE, in combination with samples containing moisture, makes the method more efficient since water in the matrix acts as a cosolvent [35]. The water in the matrix absorbs microwave energy, and cell disruption is promoted via internal superheating, which facilitates the desorption of chemicals from the matrix, thus improving the recovery of bioactive compounds, while at the same time, the mass transfer through diffusion inside the solid is increased, thus shortening the extraction time [36].

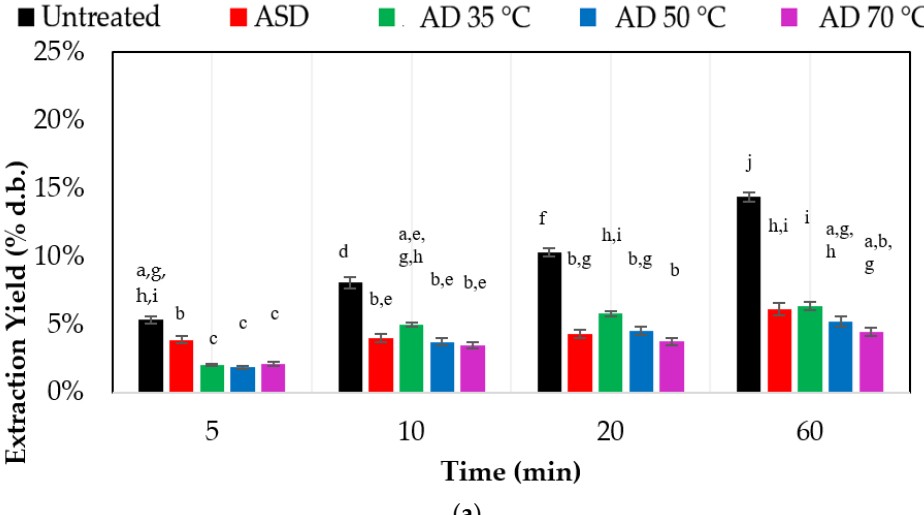

(a)

**Figure 1.** *Cont.*

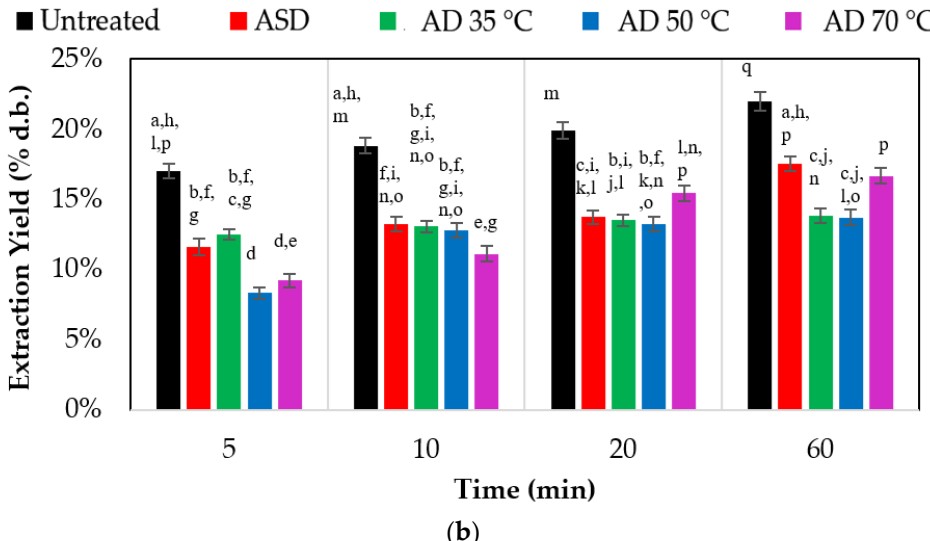

**Figure 1.** MAE extraction yields (% dry biomass, d.b.) of SOMW samples extracted using (**a**) water and (**b**) methanol as solvents. Values with different superscripts are significantly different ($p < 0.05$).

As far as MAE using methanol is concerned (Figure 1b), the untreated SOMW samples once again presented the highest *EY* (up to $21.96 \pm 0.66\%$), with *EY* values being significantly higher even in the first 5 min of extraction. For the dried samples, the highest *EY* was derived from ASD samples ($17.54 \pm 0.53\%$), followed by AD at 70 °C ($16.68 \pm 0.58\%$), indicating that drying under intense conditions can lead to structural changes that enhance methanol penetration. Moreover, comparing water and methanol, the latter can recover a wider range of compounds, both polar and nonpolar, leading to higher *EYs*. The yield between the different extraction times exhibited low differences after 10 min of extraction, showing that the combination of MAE with polar solvents in the case of dried matrices leads to effective recovery in a relatively short time.

3.2.2. Ultrasound-Assisted Extraction (UAE)

Regarding UAE, the results of the *EY* using water and methanol as solvents are presented in Figure 2a,b, respectively.

Comparing the untreated and dried SOMW samples in Figure 2a, significant differences were observed ($p < 0.05$). Similar to the case with MAE using water, the untreated SOMW samples exhibited the highest *EY*, with values equal to $29.01 \pm 0.58\%$ after 60 min of extraction. Between the different temperatures of AD, no significant differences were observed ($p > 0.05$) with the temperature of 70 °C, leading to an *EY* of $14.81 \pm 0.74\%$. In addition, after 10 min of extraction, the yield of samples with low moisture content was not affected, as also occurred in the case of MAE.

Regarding UAE using methanol (Figure 2b), a fluctuation between the *EYs* was observed for the untreated and dried SOMW. The untreated SOMW samples exhibited the highest *EY*, equal to $48.55 \pm 0.97\%$ after 60 min of extraction, followed by the samples dried using ASD ($42.46 \pm 0.98\%$) and AD 70 °C ($39.16 \pm 0.98\%$). This shows that the structural changes occurring during intense drying conditions positively affected the yield, while mild temperature treatment (AD 35 °C) led to the lowest *EY* ($19.79 \pm 0.59\%$).

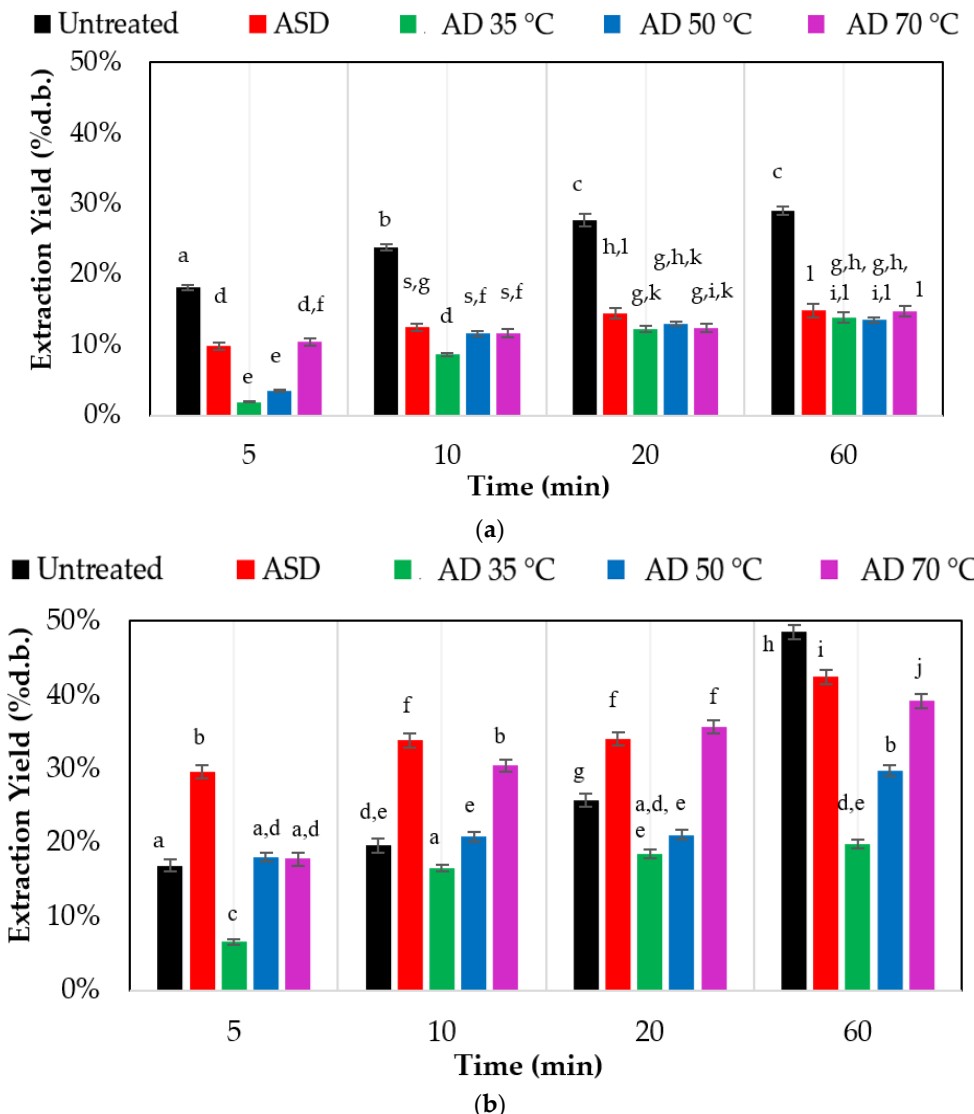

**Figure 2.** UAE extraction yields (% dry biomass, d.b.) of SOMW samples extracted using (**a**) water and (**b**) methanol as solvents. Values with different superscripts are significantly different ($p < 0.05$).

The fact that the extracts from untreated SOMW samples exhibited the highest yield for both water and methanol indicates that moisture works in conjunction with polar solvents, thus enhancing the extraction of the desirable compounds [37]. Moreover, high *EY*s were observed for both solvents after 10 min of extraction, proving that both solvents could penetrate and recover the bioactive compounds in reduced times.

Comparing MAE and UAE, it was revealed that the second method exhibited higher yields regardless of the solvent used. It is known that the mechanical effects of ultrasounds result in the greater penetration of the solvent into the cellular matrix in a short time, thus improving the mass transfer and recovery of bioactive compounds [31,38,39].

### 3.2.3. Conventional Extraction (CE) and Soxhlet Extraction (SE)

The *EYs* of the untreated and dried samples extracted with CE using water and methanol are shown in Figure 3a, while the respective *EYs* of SE are shown in Figure 3b.

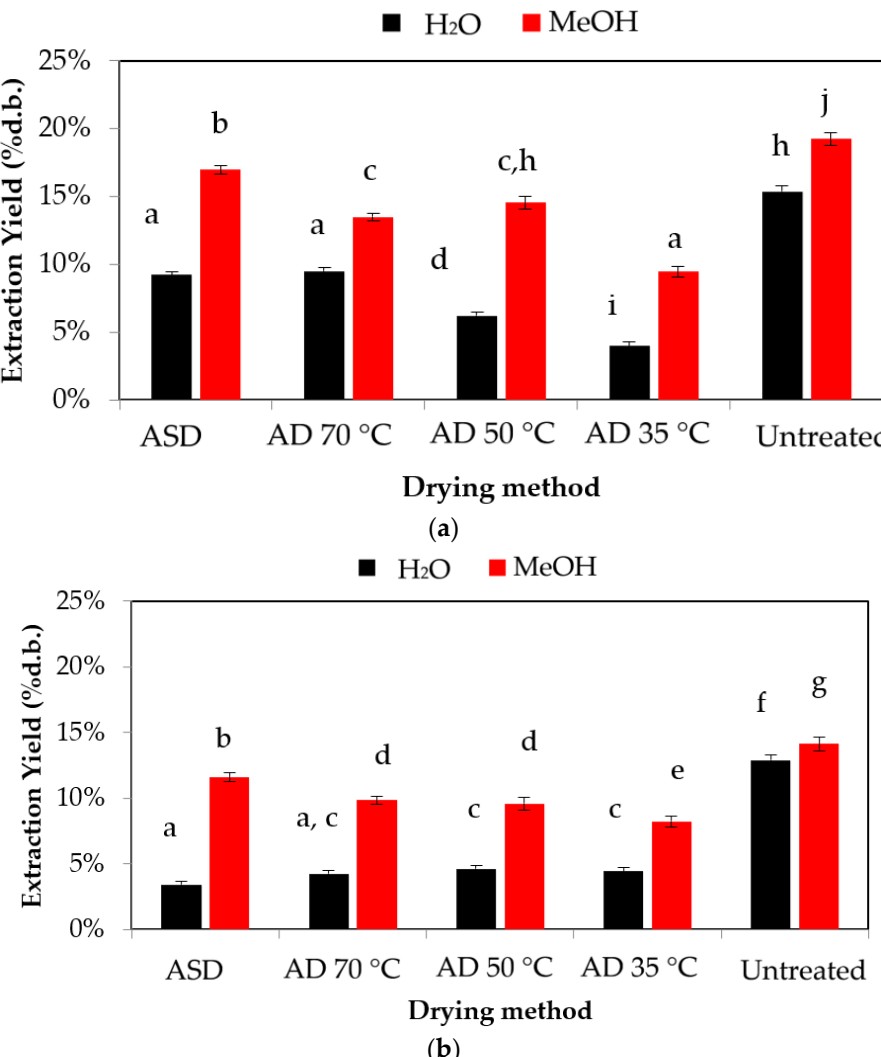

**Figure 3.** (**a**) CE and (**b**) SE extraction yields (% dry biomass, d.b.) of dried and untreated SOMW samples. Values with different superscripts are significantly different ($p < 0.05$).

Concerning the CE method using water (Figure 3a) and methanol (Figure 3b), the untreated SOMW samples had the best performance, with *EY* equal to $15.38 \pm 0.43\%$ and $19.25 \pm 0.54\%$, respectively, after 24 h of treatment, while the extracts recovered from AD 35 °C mass showed the lowest *EY* in both cases. The low yield of AD 35 °C in both cases shows that lower drying temperatures create a more uniform moisture distribution, inducing less internal stresses that allow the sample to continue to shrink until the last stages of drying [40]. Such a substantial shrinkage results in low porosity, which can affect the rehydration step and therefore the extraction itself. Furthermore, the difference between the *EY*s of the two solvents is due to their wetting ability and viscosity. Methanol, having a lower viscosity value, can more easily penetrate the plant tissue and presents higher yields regardless of the pretreatment of the sample, in contrast to water, which is a more viscous solvent.

As far as SE with water and methanol is concerned, the trend of the untreated SOMW samples presenting the best *EY* values was observed once again, while no significant differences were observed ($p > 0.05$) between the different drying methods.

Comparing CE and SE, we found that the former method exhibited slightly higher *EY*s for both selected solvents, showing that longer extraction times for conventional treatments can be more effective. However, both techniques presented significantly lower yields than MAE and UAE, with UAE being the best-performing technique. The mechanical effects of

sonication seem to decrease the reliance on the solvent itself, improving the desorption of chemicals from the matrix and increasing the yield, especially when using methanol, in which more compounds are soluble [38].

### 3.3. Antiradical Activity (AAR)

The preliminary TLC screening for the evaluation of the antioxidant capability showed that, primarily, the extracts recovered in high extraction times (60 min) exhibited the highest $A_{AR}$ and were subsequently evaluated using the DPPH assay. The results of the assay are presented in Table 2a,b.

**Table 2.** Antiradical activity of untreated and dried samples of SOMW extracts of (a) MAE and UAE methods and (b) CE and SE expressed in $IC_{50}$ values (mg/mL). The symbol $\pm$ shows the standard deviation between the replicates. Values with different superscripts are significantly different ($p < 0.05$).

| (a) | | | | |
|---|---|---|---|---|
| | **$IC_{50}$ (mg/mL)** | | | |
| | **MAE** | | **UAE** | |
| **Sample** | **Water** | **Methanol** | **Water** | **Methanol** |
| ASD | $2.45 \pm 0.02$ [i] | $2.88 \pm 0.02$ [vi] | $4.66 \pm 0.03$ [a] | $9.73 \pm 0.02$ [f] |
| AD 35 °C | $1.02 \pm 0.01$ [ii] | $5.48 \pm 0.03$ [vii] | $3.27 \pm 0.01$ [b] | $9.88 \pm 0.04$ [g] |
| AD 50 °C | $2.66 \pm 0.02$ [iii] | $5.24 \pm 0.03$ [viii] | $3.77 \pm 0.02$ [c] | $9.17 \pm 0.03$ [h] |
| AD 70 °C | $1.93 \pm 0.01$ [iv] | $3.53 \pm 0.02$ [ix] | $4.10 \pm 0.02$ [d] | $1.46 \pm 0.01$ [e] |
| Untreated | $5.60 \pm 0.03$ [v] | $2.79 \pm 0.01$ [x] | $1.40 \pm 0.01$ [e] | $2.35 \pm 0.01$ [i] |

| (b) | | | | |
|---|---|---|---|---|
| | **$IC_{50}$ (mg/mL)** | | | |
| | **CE** | | **SE** | |
| **Sample** | **Water** | **Methanol** | **Water** | **Methanol** |
| ASD | $2.23 \pm 0.02$ [i] | $6.21 \pm 0.04$ [v] | $1.10 \pm 0.01$ [a] | $4.53 \pm 0.03$ [d] |
| AD 35 °C | $5.36 \pm 0.03$ [ii] | $3.91 \pm 0.03$ [vi] | $1.02 \pm 0.01$ [b] | $5.54 \pm 0.04$ [e] |
| AD 50 °C | $3.62 \pm 0.02$ [iii] | $2.64 \pm 0.02$ [vii] | $1.00 \pm 0.01$ [b] | $4.43 \pm 0.03$ [f] |
| AD 70 °C | $3.60 \pm 0.02$ [iii] | $5.27 \pm 0.03$ [viii] | $0.74 \pm 0.01$ [c] | $1.53 \pm 0.01$ [g] |
| Untreated | $7.89 \pm 0.04$ [iv] | $3.84 \pm 0.02$ [vi] | $1.02 \pm 0.01$ [b] | $1.36 \pm 0.01$ [h] |

In general, ASD forms a rigid external layer in the dried material. Therefore, the penetration of solvent is more difficult [20]. UAE through the cavitation phenomena can destroy this rigid layer, thus allowing for the effective penetration of the solvent [41]. Therefore, in the case of ASD, the highest antioxidant capacity was observed in UAE extraction, especially when methanol is used as the solvent, since methanol is more versatile and can extract a wider range of compounds, including both polar and nonpolar antioxidants.

Examining each extraction technique separately revealed that, in the case of MAE, the dried samples exhibited a higher $A_{AR}$ than the untreated samples. This indicates that drying may cause stress to the cells and lead to the production of secondary metabolites with antioxidant activity.

In the case of MAE with methanol, the ASD method exhibited high $A_{AR}$, while in the case of AD, the samples dried at high temperatures had better $A_{AR}$ than those dried under mild temperatures. MAE using water extracts led to better antioxidant activity than MAE using methanol, especially in the case of the dried SOMW samples; however, methanol exhibited good $A_{AR}$ in the untreated samples.

Concerning UAE using water, the untreated SOMW extracts showed the highest $A_{AR}$, while in UAE using methanol, the AD 70 °C extracts had the best $A_{AR}$, followed by the untreated SOMW ones. Therefore, high antioxidant activity was achieved when SOMW was dried under elevated temperatures or not dried at all. The aqueous extracts exhibited better $A_{AR}$ than the methanolic extracts for both dried and untreated samples, while the

comparison of MAE and UAE revealed that the first method yielded extracts with higher antioxidant activity, especially when using water as the solvent. Water is considered a suitable solvent for the extraction of antioxidant compounds, which indicates that most of the bioactive compounds are polar and water-soluble.

Regarding CE (Table 2b), the extracts recovered from the samples dried with ASD and AD at 50 and 70 °C (high temperatures) showed higher $A_{AR}$ than the samples dried with AD at 35 °C (mild temperature) and untreated samples. This fact shows that drying enhanced the antioxidant activity of the SOMW samples.

In addition, SE water extracts exhibited high $A_{AR}$ with similar $IC_{50}$ values, while SE methanol extracts showed lower $A_{AR}$. Among the extracts, those obtained from untreated SOMW samples had the lowest $IC_{50}$ values, followed by AD 70 °C samples, indicating that no pretreatment or drying of the SOMW samples under high temperatures resulted in extracts with high antiradical activity.

As observed when trying to correlate the *EY* with the $A_{AR}$, despite the fact that some extracts had a high yield, they exhibited low $A_{AR}$, which may be justified by the fact that they were used to extract compounds with no antiradical activity. Between water and methanol, water appeared to be a more selective solvent for the extraction of compounds with high $A_{AR}$, and the moisture in the matrix of SOMW samples enhanced the release of bioactive compounds, which acted as a co-solvent. This did not occur in the case of methanol, where the inherent moisture can be antagonistic to the selectivity of methanol.

The SOMW samples extracted with the SE method exhibited the best antiradical activity, in comparison with MAE, UAE, and CE methods. However, SE is a time-consuming technique and may lead to the thermal degradation of phenols [29].

### 3.4. Total Phenolic Content

The total phenolic content of the SOMW extracts was determined using the Folin–Ciocalteu assay. The results of the final extracts at high extraction times are shown in Table 3a,b.

**Table 3.** The total phenolic content (TPC) ($mg_{GAE}/g_{dry\ extract}$) of untreated and dried samples of SOMW extracts of (a) MAE and UAE and (b) CE and SE methods. The symbol $\pm$ shows the standard deviation between the replicates. Values with different superscripts are significantly different ($p < 0.05$).

| (a) | | | | |
|---|---|---|---|---|
| **TPC ($mg_{GAE}/g_{dry\ extract}$)** | | | | |
| | **MAE** | | **UAE** | |
| **Sample** | **Water** | **Methanol** | **Water** | **Methanol** |
| ASD | $120.53 \pm 3.98$ [i] | $101.39 \pm 3.14$ [v] | $76.76 \pm 2.31$ [a] | $46.12 \pm 1.98$ [d] |
| AD 35 °C | $141.63 \pm 4.68$ [ii] | $89.04 \pm 2.85$ [vi] | $200.27 \pm 3.52$ [b] | $46.39 \pm 2.02$ [d] |
| AD 50 °C | $222.85 \pm 5.47$ [iii] | $58.22 \pm 2.06$ [vii] | $83.00 \pm 3.45$ [a,c] | $46.48 \pm 1.74$ [d] |
| AD 70 °C | $132.58 \pm 4.32$ [ii] | $76.75 \pm 2.34$ [viii] | $89.21 \pm 2.74$ [c] | $57.96 \pm 2.43$ [e] |
| Untreated | $177.82 \pm 5.01$ [iv] | $54.15 \pm 1.59$ [vii] | $89.60 \pm 2.83$ [c] | $106.53 \pm 3.96$ [f] |

| (b) | | | | |
|---|---|---|---|---|
| **TPC ($mg_{GAE}/g_{dry\ extract}$)** | | | | |
| | **CE** | | **SE** | |
| **Sample** | **Water** | **Methanol** | **Water** | **Methanol** |
| ASD | $108.00 \pm 3.57$ [i] | $62.90 \pm 2.37$ [v] | $186.29 \pm 4.82$ [a] | $64.50 \pm 2.87$ [e] |
| AD 35 °C | $144.33 \pm 4.23$ [ii] | $180.24 \pm 3.68$ [vi] | $230.79 \pm 5.71$ [b] | $155.02 \pm 3.94$ [f] |
| AD 50 °C | $104.41 \pm 3.29$ [i] | $301.25 \pm 6.21$ [vii] | $532.69 \pm 7.23$ [c] | $101.90 \pm 3.27$ [g] |
| AD 70 °C | $85.85 \pm 2.97$ [iii] | $50.63 \pm 1.63$ [viii] | $464.31 \pm 6.96$ [d] | $145.85 \pm 4.56$ [f] |
| Untreated | $165.42 \pm 4.74$ [iv] | $216.85 \pm 5.24$ [ix] | $185.63 \pm 4.85$ [a] | $145.69 \pm 4.44$ [f] |

Starting with MAE water extracts, those obtained from AD 50 °C samples exhibited the highest phenolic content, followed by the extracts obtained from the untreated samples. An interesting observation is that, while ASD and AD 50 °C were performed under the same temperature, the ASD extracts had a significantly lower TPC content than those of AD 50 °C. This indicates that, under solar radiation, the oxidation of phenolic compounds occurs [20,42]. A good correlation between $A_{AR}$ and TPC results was observed. In the case of methanolic MAE extracts, ASD extracts exhibited the highest phenolic content, with a good correlation with the $A_{AR}$ values, in contrast to the extracts from untreated SOMW samples, which presented the highest $A_{AR}$ but the lowest TPC. In general, water is considered more efficient than methanol since it offers extracts with high bioactive content in both TPC and $A_{AR}$.

UAE extracts presented lower TPC, which is in correlation with their $A_{AR}$. In UAE, drying under a lower temperature in combination with the use of water as solvent enhanced the release of phenolic molecules compounds, whereas the ASD extracts presented the lowest TPC, as predicted according to their low $IC_{50}$ value. Regarding extraction with methanol, the untreated samples exhibited the highest TPC, proving again that their inherent moisture acts as a cosolvent, while the dried samples presented significantly lower TPC. This is in agreement with the calculated $IC_{50}$ values, which were extremely high, indicating that UAE using methanol resulted in the recovery of extracts with low bioactive content.

Regarding the methods of CE and SE, between methanol and water, the former is considered a better solvent for the extraction of phenolic compounds, whereas comparing the different drying temperatures, it was observed that as the drying temperature increased, the phenolic content of the samples decreased. Therefore, the untreated or mildly pretreated SOMW samples presented the highest TPC.

Comparing the TPC results for MAE and UAE revealed that MAE was more effective than UAE for both dried and untreated samples, especially when using water as the solvent. In the case of the conventional extraction methods (CE and SE), SE is more efficient, especially when using water as the solvent. Moreover, the extracts from the SE method exhibited slightly higher phenolic content than the extracts from MAE, for both solvents. However, as mentioned before, SE is a slow process that requires long extraction times (at least 3 h) using significant amounts of solvent; thus, it is essential to be replaced by MAE or UAE, which are fast and efficient extraction methods.

It is also worth mentioning that the TPC and $A_{AR}$ of the extracts were influenced by the extraction temperature. Consequently, MAE and SE methods, which were conducted at higher temperatures than UAE and CE, exhibited higher bioactive content. This indicates that the diffusion of phenolic compounds in the solvent for some SOMW samples was positively affected by the temperature, maybe due to the fact that an increase in temperature lowers the viscosity of the solvent and improves the extraction rate [31]. However, the temperature must always be monitored to prevent thermal degradation, for instance, through the oxidation of the phenolic compounds.

In addition, water appeared to be the best-extracting solvent since it provided extracts rich in compounds with high antiradical activity and phenolic content, and this is in accordance with the previous research of Ramos et al. (2013) [30]. Comparing the TPC of untreated and dried samples, it was revealed that drying in combination with the right extraction method and solvent can lead to significantly higher TPC. Therefore, the right combination of drying treatment, solvent selection, and extraction technique can yield multifunctional extracts with high bioactive content.

### 3.5. Extraction Kinetics of Total Polyphenols

The recovery profile of the total phenolic compounds during MAE and UAE extractions was further studied by examining the kinetics of the recovery of the phenolic compounds. It is known the extraction process is influenced by two fundamental factors: the equilibrium and the mass transfer rate [43]. Since extraction represents a critical step

in the isolation and recovery of bioactive compounds, the mathematical modelling of the processes is essential in order to reduce energy, time, and chemical reagent consumption. The equations of Peleg's model [44] were fitted to the experimental data of MAE and UAE processes for all SOMW samples, and the model parameters were estimated. The influence of each drying method on solid–liquid extraction kinetics for MAE and UAE techniques can be seen in Figures 4 and 5.

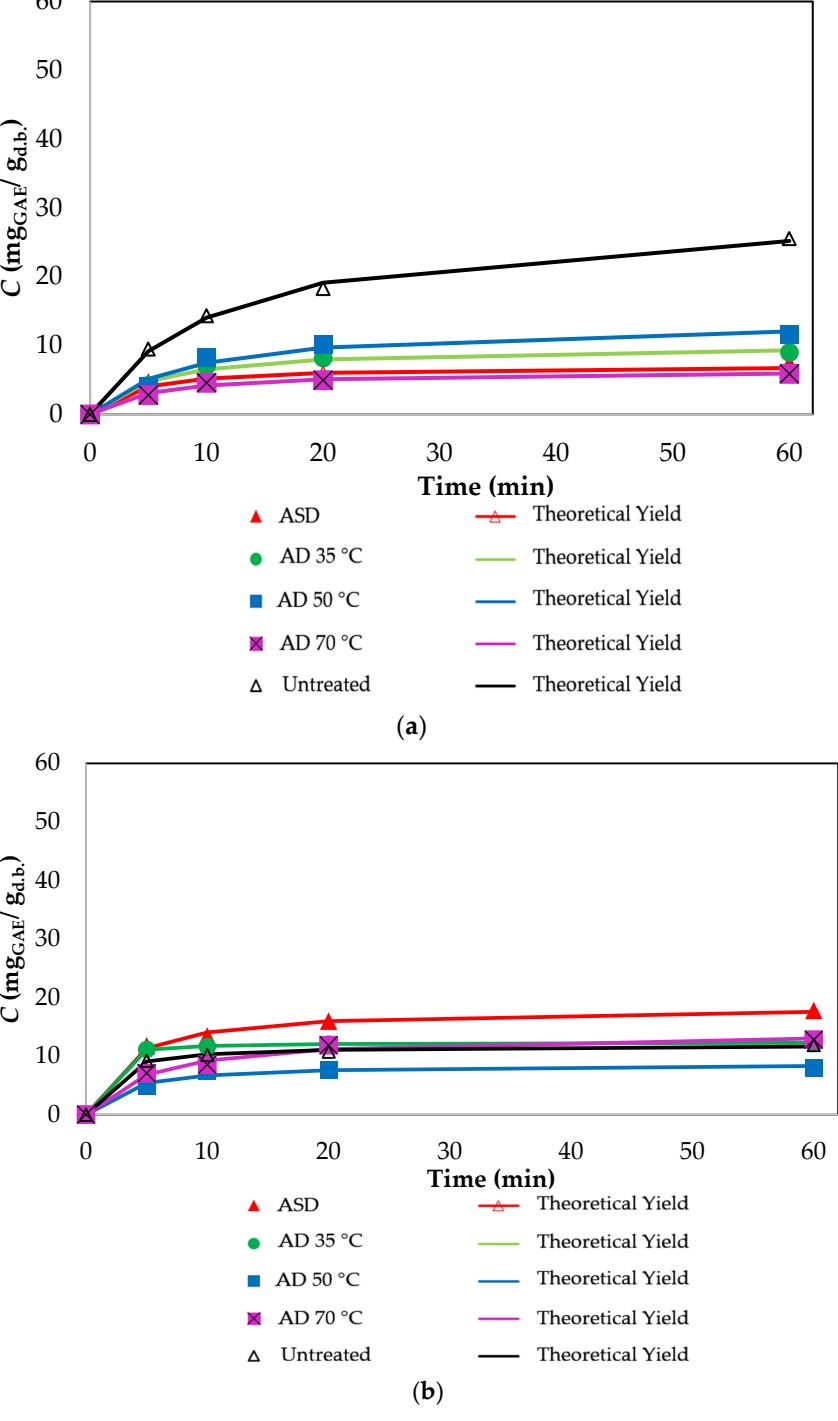

**Figure 4.** The influence of each drying method on total polyphenols of MAE extraction using (**a**) water and (**b**) methanol as solvent (symbols—experimental data; lines—approximation curves).

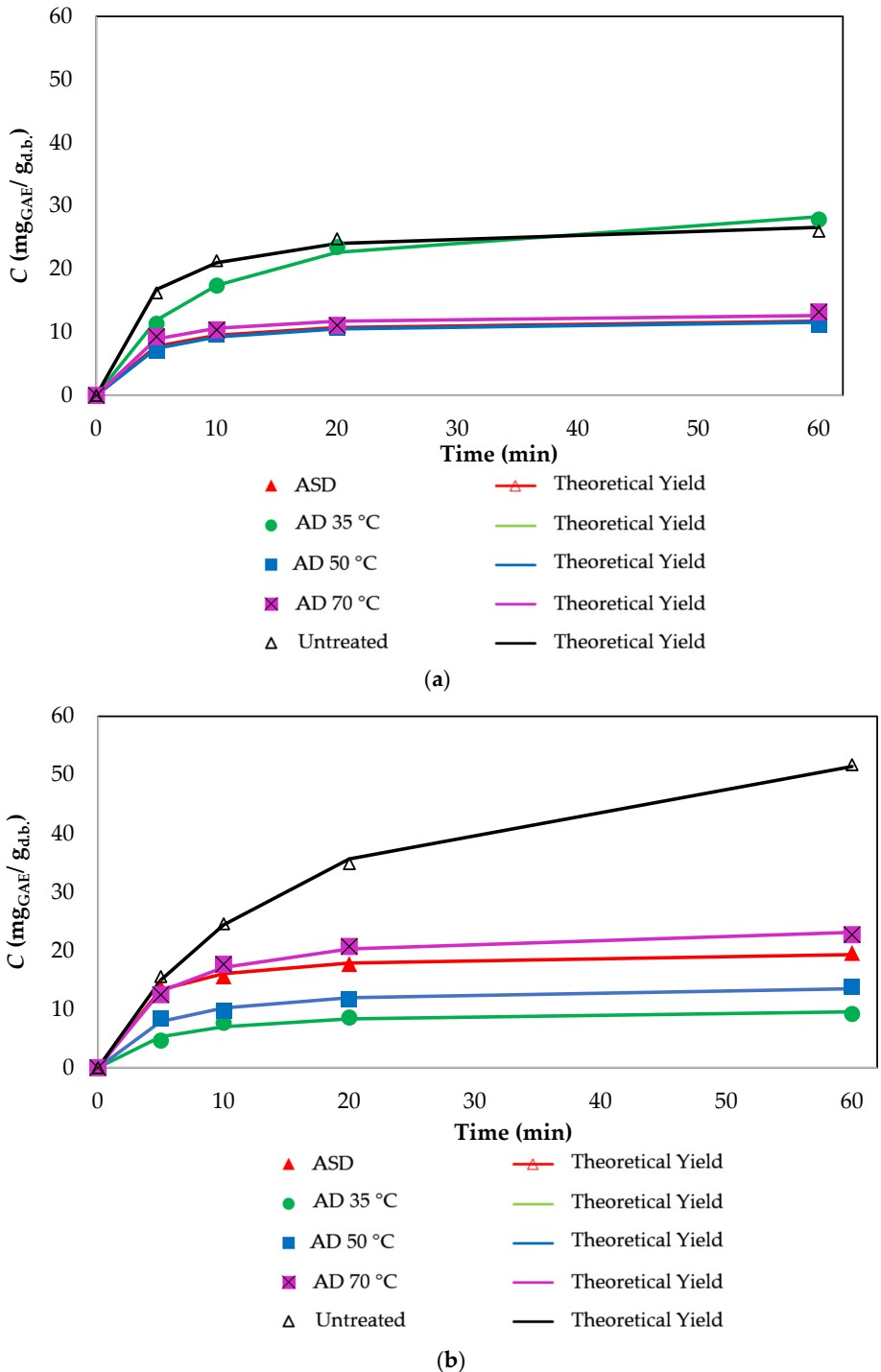

**Figure 5.** The influence of each drying method on total polyphenols of UAE extraction using (**a**) water and (**b**) methanol as solvent (symbols—experimental data; lines—approximation curves).

All the extraction curves of Figures 4 and 5 present a characteristic extraction profile indicating a high initial rate of polyphenol extraction at the beginning of the process, followed by a slower extraction rate, which leads to the asymptotical approach of the equilibrium concentration.

The highest concentration of the extracted polyphenols was achieved during the UAE of untreated SOMW samples using methanol at 60 min, whereas the lowest was during MAE using water for the extraction of SOMW samples dried with AD at 70 °C. When comparing MAE and UAE, differences in the initial extraction rate were observed.

MAE was expected to present a higher initial extraction rate than UAE since the elevated temperature of MAE (50 °C) would accelerate the mass transfer and dissolution of the bioactive compounds because of the decrease in the viscosity of the solvent. However, that is not always the case because the rate is also affected by the structure of the material. At higher temperatures, the equilibrium of bioactive compound dissolution is reached in a short processing time, and the bioactive compounds released in the solute can gradually degrade, leading to lower maximum concentrations [45,46].

The best performance was indeed at 60 min of extraction, but small differences were observed with lower extraction times, especially 20 min. After 20 min, a plateau was reached in the majority of the cases. With the aim of energy consumption minimisation and the sustainability of our processes, lower extraction times were preferred since they resulted in approximately the same extraction yields as higher extraction times.

Moreover, the rate of the extraction is also affected by the properties of the matrix, with many dried samples presenting a low wetting rate and, thus, a lower TPC content, depending on the drying conditions.

As can be seen, the results of Peleg's model, which was used to describe the kinetics of the UAE and MAE, were found to be compatible with the experimental data. This good compatibility can be better certified by the high correlation coefficient values (q) shown in Table 4. This table also includes the calculated parameters of Peleg's model (constants $k_1$ and $k_2$) and the root-mean-squared deviation (RMSD).

**Table 4.** Peleg's rate constant $K_1$ (min $g_{db}/mg_{GAE}$), Peleg's capacity constant $K_2$ ($g_{db}/mg_{GAE}$), extraction rate at the very beginning $B_0$ ($mg_{GAE}/g_{db}$ min, equilibrium yield concentration of total extracted polyphenols ($mg_{GAE}/g_{db}$) ($C_e$), correlation coefficient (q), and root-mean-squared deviation (RMSD) using different extraction and drying methods.

| Extraction Method | Drying Method | $K_1$ (min·$g_{db}/mg_{GAE}$) | $K_2$ ($g_{db}/mg_{GAE}$) | $B_0$ ($mg_{GAE}/g_{db}$·min) | $C_e$ ($mg_{GAE}/g_{db}$) | q | RMSD |
|---|---|---|---|---|---|---|---|
| | ASD | 0.53 | 0.14 | 1.87 | 7.17 | 0.97 | 0.99 |
| | AD 35 °C | 0.55 | 0.10 | 1.83 | 10.19 | 0.99 | 1.00 |
| MAE H$_2$O | AD 50 °C | 0.61 | 0.07 | 1.64 | 13.74 | 0.99 | 0.99 |
| | AD 70 °C | 0.84 | 0.15 | 1.19 | 6.49 | 0.99 | 1.00 |
| | Untreated | 0.38 | 0.03 | 2.64 | 29.97 | 1.00 | 1.00 |
| | ASD | 0.17 | 0.05 | 0.35 | 18.49 | 1.00 | 1.00 |
| | AD 35 °C | 0.05 | 0.08 | 20.96 | 12.41 | 1.00 | 1.00 |
| MAE MeOH | AD 50 °C | 0.35 | 0.11 | 2.83 | 8.76 | 1.00 | 0.99 |
| | AD 70 °C | 0.38 | 0.07 | 2.63 | 14.20 | 1.00 | 1.00 |
| | Untreated | 0.13 | 0.08 | 7.62 | 11.93 | 1.00 | 1.00 |
| | ASD | 0.24 | 0.08 | 4.19 | 12.31 | 1.00 | 1.00 |
| | AD 35 °C | 0.27 | 0.03 | 3.76 | 32.35 | 1.00 | 1.00 |
| UAE H$_2$O | AD 50 °C | 0.26 | 0.08 | 3.81 | 12.19 | 1.00 | 1.00 |
| | AD 70 °C | 0.18 | 0.08 | 5.53 | 13.14 | 1.00 | 1.00 |
| | Untreated | 0.12 | 0.04 | 8.29 | 28.12 | 1.00 | 1.00 |
| | ASD | 0.13 | 0.05 | 7.89 | 20.13 | 1.00 | 1.00 |
| | AD 35 °C | 0.46 | 1.00 | 2.17 | 10.29 | 0.99 | 1.00 |
| UAE MeOH | AD 50 °C | 0.29 | 0.07 | 3.48 | 14.40 | 1.00 | 1.00 |
| | AD 70 °C | 0.18 | 0.04 | 5.50 | 24.85 | 1.00 | 1.00 |
| | Untreated | 0.26 | 0.02 | 3.86 | 66.09 | 1.00 | 1.00 |

$K_1$ is Peleg's rate constant (min·$g_{db}/mg_{GAE}$), which expresses the rate of the TPC extraction and is inversely related to the initial TPC extraction rate at the very beginning ($B_0$). On the other hand, $K_2$ is Peleg's capacity constant ($g_{db}/mg_{GAE}$) and is related to the maximum attainable TPC content [44]. In most cases, $B_0$ was higher for the untreated samples than for dried samples. This can be explained by the two-stage extraction of phenolics, whereby the first stage encompasses the dissolution of the phenolics around the matrix surface (washing), which proceeds very rapidly, while the second stage encompasses the slow diffusion of the phenolics from the matrix to the solvent.

In addition, in most cases, the maximum extraction capacity ($C_e$) was higher for the untreated samples than for the dried samples. Higher values of $C_e$ imply more extractable compounds. Therefore, the present results revealed that processes in elevated temperatures

damaged the heat-sensitive phenolic compounds, whereas the application of ultrasounds with untreated samples or samples dried with gentle drying methods corresponded to the highest $C_e$. This implies that the cavitation caused by UAE could disrupt the structure of untreated samples, enable more target compounds exposed to solvent, and break the bonds between target compounds.

Finally, the coefficient of determination for the predicted total phenolic extraction yields showed a good correlation with the experimental data since the q values varied from 0.97 to 1.00, and RMSD was higher than 0.99. This indicates that the nonexponential Peleg's model can be employed to predict the extraction of phenolics during UAE and MAE.

### 3.6. Identification of Phenolic Compounds

The extracts with the most promising antioxidant potential and total phenolic content, namely MAE water extracts from untreated and AD 50 °C SOMW mass, were analysed with HPLC-DAD in order to identify their main metabolites. The chromatographic profiles of the extracts acquired using HPLC-DAD at 280, 320, and 365 nm are presented in Figure 6.

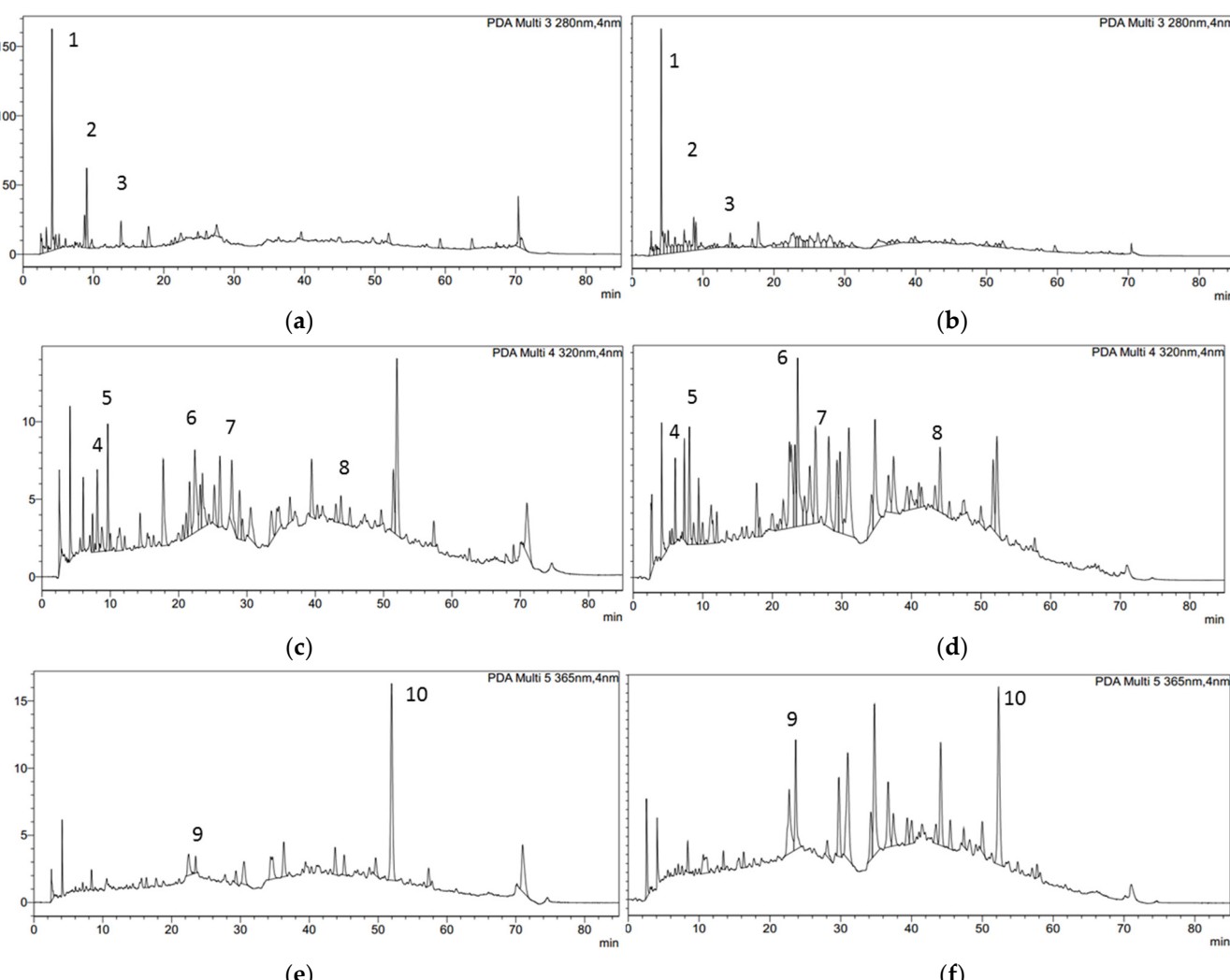

**Figure 6.** HPLC-DAD chromatographs of SOMW untreated (left, (**a**,**c**,**e**)) and air-dried at 50 °C (right, (**b**,**d**,**f**)) MAE water extracts monitored at 280 nm (**a**,**b**), 320 nm (**c**,**d**), and 365 nm (**e**,**f**). Estimated peak assignment: 1: hydroxytyrosol glucoside, 2: hydroxytyrosol, 3: hydroxytyrosol derivative, 4: elenolic acid glucoside, 5: dimethyloeleuropein; 6: dihydro-oleuropein; 7: oleuropein, 8: oleuropein aglycone isomer, 9: 3,4-DHPEA-EDA, 10: luteolin.

The chromatograms revealed that SOMW is a rich source of several phenolic compounds. Among the different compounds, hydroxytyrosol was identified with a reference standard solution, while the other compounds were identified by comparing the retention times and shape of the HPLC spectra of the chromatographic peaks with similar literature data [47–50].

The majority of the identified compounds exhibited λmax at 278 and 236 nm, 242 and 238 nm, 281 and 244 nm, and 258 and 355 nm, which are quite similar to hydroxytyrosol (λmax = 280 and 240 nm), elenolic acid glucoside (λmax = 244 and 236 nm), oleuropein (λmax = 280 and 246 nm), and luteolin (λmax = 255 and 353 nm), respectively, suggesting that both samples presented quite a few derivatives of hydroxytyrosol, elenolic acid, oleuropein, and luteolin. Hydroxytyrosol and hydroxytyrosol derivatives were identified at 280 nm, whereas some flavonoid compounds, such as luteolin, were found at 365 nm. This is in agreement with the study of Mulinacci et al. and Gomez-Cruz et al. [47,49]. The secondary metabolites from *Olea europea* L. have high biological value, and they are present in different concentrations in the various parts of the olive plant, while flavonoids are the most important phytochemical compounds present in many plants, fruits, and vegetables, providing many health benefits [51,52].

Starting with hydroxytyrosol, the majority of hydroxytyrosol glucoside (peak 1), which is the primary form present in olive fruits, degrades to hydroxytyrosol (peak 2) during malaxation, and therefore it is abundant in all olive-derived matrices (paste, pomace, wastewater, and oil). The compound eluting at 7.9 min presented UV–Vis spectral characteristics closely related to that of elenolic acid glucoside (peak 4), while at 10.4 min, dimethyloleuropein seemed to be the most fitting compound (peak 5). Dimethyloleuropein is also known as the oleuropein glucosylated derivative formed during olive fruit maturation [53]. The peak numbers 6–8 present several similarities with oleuropein and oleuropein derivatives, such as dihydro-oleuropein and oleuropein aglycone. These compounds were detected in several olive fruit by-products [54–56], since they are a degrading form of olive fruits' secoiridoids, i.e., oleuropein and dimethyloleuropein, which occur during crushing/malaxation, thus forming several secoiridoid aglycone derivatives [25,57]. Moreover, fruit flavonoids such as luteolin-7-O-glucoside and rutin, as well as cinnamic acid esters such as verbascoside and comselogoside, have been reported in fruit derivatives (paste, pomace, and wastewater) [25], indicating their preferred transfer to waste without many alterations. Finally, luteolin derivatives were also detected in the chromatograph of the extracts at 365 nm. All the discussed chemical compounds are presented in Table 5. Hydroxytyrosol, hydroxytyrosol glucoside, and oleuropein aglycone isomer were also found in the study of Spizziri et al. [58].

**Table 5.** Identified compounds of SOMW untreated and air-dried at 50 °C MAE water extracts using HPLC-DAD.

| Peak Number | Identified Compounds | Retention Time (min) | Monitored Wavelength (nm) |
|---|---|---|---|
| 1 | Hydroxytyrosol glucoside | 4.1 | 280 |
| 2 | Hydroxytyrosol | 8.7 | 280 |
| 3 | Hydroxytyrosol derivative | 13.9 | 280 |
| 4 | Elenolic acid glucoside | 7.9 | 320 |
| 5 | Dimethyloeleuropein | 10.4 | 320 |
| 6 | Dihydro-oleuropein | 22.3 | 320 |
| 7 | Oleuropein | 26.0 | 320 |
| 8 | Oleuropein aglycone isomer | 44.2 | 320 |
| 9 | 3,4-DHPEA-EDA | 22.7 | 365 |
| 10 | Luteolin | 52.1 | 365 |

The differences in the number of peaks in the chromatographs of the untreated and dried SOMW extracts showed that the conversion of some compounds to different phenolic derivatives was mainly due to thermal treatment. However, the total phenolic content of

AD 50 °C was significantly higher than that of the untreated SOMW, as also validated in the chromatographs of 320 and 365 nm. All the discussed chemical compounds are presented in Table 5.

## 4. Conclusions

Solid olive mill waste can be considered a high-value by-product thanks to its high phenolic content, especially hydroxytyrosol and oleuropein derivatives, which have been shown to have biological and antioxidant activity. Microwave- and ultrasound-assisted extraction using water or methanol as solvents were applied for the recovery of the desirable compounds from both untreated and dried samples. Methanol extracts presented a higher yield, while extraction methods using water were found to be more selective in the recovery of phenolic compounds. Comparing untreated and dried samples at different temperatures, it was revealed that mild treatment temperatures in combination with microwave and ultrasound extraction techniques resulted in final extracts with high total phenolic content. However, the untreated samples showed high $EYs$, whereas the dried samples seemed to be more selective in terms of $A_{AR}$ and TPC. In terms of a comparison between microwave and ultrasound treatment, the former exhibited extracts richer in bioactive compounds, while the latter was considered a more effective technique in terms of yield. The effective use of water as a solvent provides an opportunity for such total extracts to be used in the food sector. One innovation in studying the extraction of phenolics from olive mill pomace involves the application of advanced extraction techniques such as ultrasound-assisted extraction (UAE) and microwave-assisted extraction (MAE). According to the results of our study, these methods were found to significantly enhance extraction efficiency, reduce extraction time, and minimise the use of organic solvents. Specifically, the use of UAE in the extraction of phenolics from olive mill pomace has shown promising results, with higher extraction yields and increased phenolic content compared with traditional methods. Moreover, UAE is suitable for industrial-scale applications, since larger ultrasound systems, such as industrial-scale sonicators or flow-through systems, are employed. These systems are designed to handle larger volumes and accommodate continuous or semi-continuous extraction processes. Industrial-scale UAE can be integrated into existing production lines, allowing for the efficient and continuous extraction of the desired compounds from raw materials, providing the possibility to design a scalable agri-food by-product valorisation pathway. Under this framework, the valorisation of olive mill pomace will be economically feasible and scalable and thus it is of interest for industrial applications and has commercial potential. In conclusion, solid olive mill waste represents a rich source of bioactive compounds with considerable antioxidant properties, and the proper drying technique in conjunction with the right extraction method and solvent can lead to the significant recovery of phenolic compounds and the valorisation of this valuable by-product stream.

**Author Contributions:** Conceptualisation, M.K. and K.K.; methodology, T.-V.M.; software, K.K.; validation, K.K. and S.P.; formal analysis, M.S., S.P. and A.A.-D.; investigation, K.K.; resources, M.K.; data curation, T.-V.M., K.K. and M.S.; writing—original draft preparation, T.-V.M. and K.K.; writing—review and editing, M.S., S.P. and A.A.-D.; visualisation, M.S. and S.P.; supervision, K.K. and M.K.; project administration, M.K.; funding acquisition, M.K. All authors have read and agreed to the published version of the manuscript.

**Funding:** This research received no external funding.

**Data Availability Statement:** The datasets used and/or analysed during the current study are available from the corresponding author at a reasonable request.

**Conflicts of Interest:** The authors declare no conflict of interest.

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
