# Peer review of "The Recovery of Bioactive Compounds from Olive Pomace Using Green Extraction Processes"

_resources, doi:10.3390/resources12070077_

Round 1

Reviewer 1 Report

Lines 1-3: inappropriate use of the term "antioxidants compounds" in the title - you are not extracting only antioxidants - please change to reflect the nature of the molecules you are extracting from olive pomace (compare with Lines 21-23)

Lines 23-24: inaccurate. Please correct this as hydroxytyrosol is already an oleuropein derivative. What about luteolin derivatives? Do you have any chromatographic confirmation?

Lines 36-44: SOMW IS NOT the only form of pomace formed in the modern production of (E)VOO. Please consider this fact and specify why you have chosen to elaborate ONLY pomace from the 3-phase systems while it is well known that 2-phase systems are dominant nowadays (alperujo)?

Lines 81-85: Can you give some more details, please? How many samples, cultivar, time between oil extraction and pomace sample(s) collection (very important!!!), etc.

Line 90: Was hydroxytyrosol the only phenols standard you used?

Line 121: Correct the spelling (grounded). What were the dimensions of the ground pomace particles?

Lines 125-126, 132-133, 137-138 & 142-143: please explain the use of different amounts - initial pomace/solvent ratios.

Lines 149-152: Do you mean "... obtained DRIED extract relative ..."?

Line 168: Please use subscripts in the formula (t, t=0).

Lines 181-185, 189 & 193: unclear use of symbols/units in the definitions of C(t), K1 & K2. Please correct.

Lines 207-210: several missing spaces in the gradient specification

Lines 202 & 211: You are using two different abbreviations for the same type of detector: PDA and DAD. Please unify.

Lines 237-238: Please explain what the tolerances express (±).

Figures 1, 2, 3, 4 & 5; Tables 1, 2 & 3: Please correct the degrees Celsius symbol use: should be written as ºC.

Figures 1, 2 & 3: Please explain the y-axis meaning & units used. Please explain the meaning of the small letters a,b, c, d ... in the graphs.

Figure 3: Please correct the water bars to read: H2O.

Tables 1 & 2: Please explain what the tolerances express (±), and please explain the meaning of the superscripts used (i, ii, iii ...)

Figures 4 & 5: typo in the explanation of the graph: correct to "Theoretical".

Line 514: Please correct to ºC.

Table 3: Please explain the meaning of the units for K1, K2, B0 Ce.

Figure 6 & Lines 561-565: If I understand well your peaks assignment was based ONLY on the absorbance maximum data and retention times comparison with similar literature data. But unfortunately, you don't cite this literature data! Please explain why. I find your peak assignment not accurate enough to be published in this way. The only standard you use is hydroxytyrosol. Please provide more confirmative evidence.

Lines 561-565: Missing spaces.

Lines 570-591: I find your conclusions inaccurate as they are based solely on the absorption and retention times data comparison with the literature WITHOUT any evidence from the standards or MS determination. Please explain or rephrase.

Author Response

Recovery of Bioactive Compounds from Olive Pomace Using Green Extraction Processes

Response to Reviewer 1

We would like to thank the Reviewer for the useful comments and suggestions made on our manuscript, which helped us improve it substantially. Please find below the changes made on our manuscript following the Reviewer’s recommendations.

-Lines 1-3: inappropriate use of the term "antioxidants compounds" in the title - you are not extracting only antioxidants - please change to reflect the nature of the molecules you are extracting from olive pomace (compare with Lines 21-23)

The title is modified to “Recovery of Bioactive Compounds from Olive Pomace Using Green Extraction Processes”.

-Lines 23-24: inaccurate. Please correct this as hydroxytyrosol is already an oleuropein derivative. What about luteolin derivatives? Do you have any chromatographic confirmation?

Thank you for this comment. Oleuropein is deleted. HPLC chormatograms are presented in Results and Discussion Section and confirm the presence of hydroxytyrosol.

-Lines 36-44: SOMW IS NOT the only form of pomace formed in the modern production of (E)VOO. Please consider this fact and specify why you have chosen to elaborate ONLY pomace from the 3-phase systems while it is well known that 2-phase systems are dominant nowadays (alperujo)?

Greece is one of four European countries that produce 12.5 million tonnes of olive oil in total, accounting for more than 95% of total EU production. The majority of olive mills operating in Greece are three-phase centrifugal https://doi.org/10.1007/978-3-030-68074-9_166-1 . A few old-style presses are also preserved. Two-phase oil mills are not very widespread in Greece. Our search aims to develop strategies of valorization of SOMW, using as case study the value chain of Greek olive oil production. Therefore, this study focused on the valorization of pomace from the 3-phase systems olive mills https://doi.org/10.1002/cbic.202200254.

-Lines 81-85: Can you give some more details, please? How many samples, cultivar, time between oil extraction and pomace sample(s) collection (very important!!!), etc.

The sample was from the cultivated tree variety Koroneiki I-38 and it was one collected and separated in smaller beakers. The oil extraction and pomace sample collection were performed in the same day but unfortunately we do not have more information about the exact time. Please let us know if more data are requested.

-Line 90: Was hydroxytyrosol the only phenols standard you used?

Hydroxytyrosol standard was used for the qualitative determination of some HPLC peaks in the obtained chromatograms, where the other peaks are identified by comparing retention times with absorption spectrum data from literature.

-Line 121: Correct the spelling (grounded). What were the dimensions of the ground pomace particles?

It is corrected. The mean dimension of the ground pomace particles was 900 μm.

Lines 125-126, 132-133, 137-138 & 142-143: please explain the use of different amounts - initial pomace/solvent ratios.

The biomass/solvent ratio for each extraction process had been optimized in previous studies of our team. Therefore, the optimum biomass/solvent ratio for each method were used in our study (https://www.chemeng.ntua.gr/the_thesis_archive, Diploma Thesis of ΜΙΣΥΡΛΗ ΘΕΟΔΩΡΑ-ΒΕΝΕΤΙΑ)

-Lines 149-152: Do you mean "... obtained DRIED extract relative ..."?

Yes, thank you for the comment. It is modified.

-Line 168: Please use subscripts in the formula (t, t=0).

Subscripts are used.

-Lines 181-185, 189 & 193: unclear use of symbols/units in the definitions of C(t), K1 & K2. Please correct.

More explanations about the symbols and their units are given.

-Lines 207-210: several missing spaces in the gradient specification

The missing spaces are corrected.

-Lines 202 & 211: You are using two different abbreviations for the same type of detector: PDA and DAD. Please unify.

It is unified.

-Lines 237-238: Please explain what the tolerances express (±).

The explanation is given.

-Figures 1, 2, 3, 4 & 5; Tables 1, 2 & 3: Please correct the degrees Celsius symbol use: should be written as ºC.

The figures are corrected.

-Figures 1, 2 & 3: Please explain the y-axis meaning & units used. Please explain the meaning of the small letters a,b, c, d ... in the graphs.

The meanings are added.

-Figure 3: Please correct the water bars to read: H2O.

They are corrected

-Tables 1 & 2: Please explain what the tolerances express (±), and please explain the meaning of the superscripts used (i, ii, iii ...)

The explanations are given.

-Figures 4 & 5: typo in the explanation of the graph: correct to "Theoretical".7

It is fixed.

-Line 514: Please correct to ºC.

It is corrected.

-Table 3: Please explain the meaning of the units for K1, K2, B0 Ce.

The explanations are given in the caption.

Figure 6 & Lines 561-565: If I understand well your peaks assignment was based ONLY on the absorbance maximum data and retention times comparison with similar literature data. But unfortunately, you don't cite this literature data! Please explain why. I find your peak assignment not accurate enough to be published in this way. The only standard you use is hydroxytyrosol. Please provide more confirmative evidence.

Thank you for this helpful comment and apologies for not including the literature data. The citations are added, a table including all the identified peaks is included and it would be worth mentioning that the comparison was not only based on the retention times and absorbance maximum, but also according to the spectral shape of the HPLC spectra of the chromatographic peaks. A better explanation and comparison with literature is provided and we hope we meet your requirements.

-Lines 561-565: Missing spaces.

They are corrected.

Lines 570-591: I find your conclusions inaccurate as they are based solely on the absorption and retention times data comparison with the literature WITHOUT any evidence from the standards or MS determination. Please explain or rephrase.

Some sentences are provided in order to better explain the results of our HPLC measurements.

Reviewer 2 Report

This manuscript is about the use of green extraction methods to recover metabolites from olive pomace and I have some considerations:

- What is the granulometry of SOMW used in the extractions?

- Why methanol was chosen instead of ethanol? I ask due to the toxicity of methanol

- How the temperature was controlled in Ultrasound-assisted extraction?

- Why the biomass/solvent ratio was different in each type of extraction?

- The SOMW lipids were removed before extractions? If not, lipids could extract phenolics instead going to water, could not?

- The identification of phenolics of each extract was realized? I think it could be interesting a discussion of the impact of green extraction methods in the degradation of these compounds.

Author Response

Recovery of Bioactive Compounds from Olive Pomace Using Green Extraction Processes

Response to Reviewer 2

We would like to thank the Reviewer for the useful comments and suggestions made on our manuscript, which helped us improve it substantially. Please find below the changes made on our manuscript following the Reviewer’s recommendations.

This manuscript is about the use of green extraction methods to recover metabolites from olive pomace and I have some considerations:

- What is the granulometry of SOMW used in the extractions?

Prior to extraction, the dried samples were ground to increase the active surface area. The mean dimension of the ground pomace particles was 900μm.

- Why methanol was chosen instead of ethanol? I ask due to the toxicity of methanol?

Methanol was chosen in the extractions instead of ethanol as methanol and water are the solvents that can extract more diversity of compounds, which can explain the high extraction efficiency. According to the literature and previous experiments in our lab, methanol shows higher bioactive compound content compared to ethanol presenting more physical and chemical advantages. In addition, it has a lower boiling point than ethanol needing lower temperature to evaporate and leading to less energy consumption https://doi.org/10.1016/j.ejpe.2015.06.007 , https://doi.org/10.3390/antibiotics9020048 , https://doi.org/10.1016/S1350-4177(00)00075-4 , https://doi.org/10.5539/ijc.v1n1p36 . However this recommendation about the use of ethanol is very useful for us and in our next experiments we will consider the additional use of ethanol as a solvent in our extractions.

- How the temperature was controlled in Ultrasound-assisted extraction?

In UAE the temperature was controlled during the extractions and remained stable at 25 °C using a coolant passing through the double-wall extraction beaker so that the risk of degrading thermos-labile components is minimised. The temperature inside the extraction beaker was measured using a thermocouple and was shown in the screen of the extraction system.

- Why the biomass/solvent ratio was different in each type of extraction?

The biomass/solvent ratio for each extraction process had been optimized in previous studies of our team. Therefore, the optimum biomass/solvent ratio for each method were used in our study (https://www.chemeng.ntua.gr/the_thesis_archive, Diploma Thesis of ΜΙΣΥΡΛΗ ΘΕΟΔΩΡΑ-ΒΕΝΕΤΙΑ)

- The SOMW lipids were removed before extractions? If not, lipids could extract phenolics instead going to water, could not?

The most important antioxidants are phenolic chemicals, which are classified as lipophilic or hydrophilic. While lipophilic phenols such as tocopherols can be found in other vegetable oils, the majority of hydrophilic phenols in olive oil are unique to the Olea europaea species, giving it chemotaxonomic significance. Therefore, we focus on the recovery of hydrophilic phenols mainly.

- The identification of phenolics of each extract was realized? I think it could be interesting a discussion of the impact of green extraction methods in the degradation of these compounds.

The extracts with the most promising antioxidant potential and total phenolic content, namely MAE water extracts from untreated and AD 50°C SOMW mass, were analyzed with HPLC-DAD in order to identify their main metabolites. This analysis would be the part of a forthcoming publication of our team since it is a topic of great interest that needs deep investigation and we consider that should be a separate research paper.

Reviewer 3 Report

Dear Authors,

this is an interesting article investigating the antioxidant chemical constituents obtained from olive pomace through green extraction processes. The manuscript and the data presented are clear and consistent. The overall English language is good. Most sections are well prepared. I would be grateful if the authors could improve the manuscript by eliminating grammar and reference typos across the whole text. 

Here are some suggestions:

Introduction: I would be grateful if the authors may delete lines 70-78.

Materials and Methods: please add the botanical name of the plant of the oil samples. Please, add the voucher specimen if possible.

I would appreciate the author adding a table in section 2.4 summarizing all the extractions performed.

Results: Line 570 I suggest the authors add a table regarding the chemical compounds discussed.

Figure 6: Please, add letters to each chromatogram.

Bibliography: please check typos.

I would suggest these recent articles on antioxidant compounds and olive oil to improve the literature section:

Limongelli, F.; Crupi, P.; Clodoveo, M.L.; Corbo, F.; Muraglia, M. Overview of the Polyphenols in Salicornia: From Recovery to Health-Promoting Effect. Molecules 2022, 27, 7954. https://doi.org/10.3390/molecules27227954

Spizzirri, U.G.; Caputo, P.; Oliviero Rossi, C.; Crupi, P.; Muraglia, M.; Rago, V.; Malivindi, R.; Clodoveo, M.L.; Restuccia, D.; Aiello, F. A Tara Gum/Olive Mill Wastewaters Phytochemicals Conjugate as a New Ingredient for the Formulation of an Antioxidant-Enriched Pudding. Foods 2022, 11, 158. https://doi.org/10.3390/foods11020158

Clodoveo, M.L.; Muraglia, M.; Crupi, P.; Hbaieb, R.H.; De Santis, S.; Desantis, A.; Corbo, F. The Tower of Babel of Pharma-Food Study on Extra Virgin Olive Oil Polyphenols. Foods 2022, 11, 1915. https://doi.org/10.3390/foods11131915

Dear authors,

the overall English is good. Some sentences are long and using tables might support the readiness of the text. Some typos are present, please read carefully the whole manuscript before publication to avoid them.

Author Response

Recovery of Bioactive Compounds from Olive Pomace Using Green Extraction Processes

Response to Reviewer 3

Dear Authors,

this is an interesting article investigating the antioxidant chemical constituents obtained from olive pomace through green extraction processes. The manuscript and the data presented are clear and consistent. The overall English language is good. Most sections are well prepared. I would be grateful if the authors could improve the manuscript by eliminating grammar and reference typos across the whole text. 

We would like to thank the Reviewer for the useful comments and suggestions made on our manuscript, which helped us improve it substantially. Please find below the changes made on our manuscript following the Reviewer’s recommendations.

Here are some suggestions:

-Introduction: I would be grateful if the authors may delete lines 70-78.

The paragraph was shortened and many technical details in these lines are removed.

-Materials and Methods: please add the botanical name of the plant of the oil samples. Please, add the voucher specimen if possible.

The olive cultivar is Koroneiki. I-38. A sentence is added in Section 2.1.

-I would appreciate the author adding a table in section 2.4 summarizing all the extractions performed.

A table describing the extraction conditions is added.

-Results: Line 570 I suggest the authors add a table regarding the chemical compounds discussed.

A table with all the identified compounds (Table 5) is added.

-Figure 6: Please, add letters to each chromatogram.

Letters in the caption of each chromatogram are added.

-Bibliography: please check typos.

Some typos are corrected. Please inform us if you would like us to correct something else. The reference management software used is Mendelay and it is still used in this stage of review because it helps in the easy modification of references. The authors will fix the references and export them as text in the final version of the manuscript, so that they are in agreement with the style of the journal.

I would suggest these recent articles on antioxidant compounds and olive oil to improve the literature section:

Limongelli, F.; Crupi, P.; Clodoveo, M.L.; Corbo, F.; Muraglia, M. Overview of the Polyphenols in Salicornia: From Recovery to Health-Promoting Effect. Molecules 2022, 27, 7954. https://doi.org/10.3390/molecules27227954

Spizzirri, U.G.; Caputo, P.; Oliviero Rossi, C.; Crupi, P.; Muraglia, M.; Rago, V.; Malivindi, R.; Clodoveo, M.L.; Restuccia, D.; Aiello, F. A Tara Gum/Olive Mill Wastewaters Phytochemicals Conjugate as a New Ingredient for the Formulation of an Antioxidant-Enriched Pudding. Foods 2022, 11, 158. https://doi.org/10.3390/foods11020158

Clodoveo, M.L.; Muraglia, M.; Crupi, P.; Hbaieb, R.H.; De Santis, S.; Desantis, A.; Corbo, F. The Tower of Babel of Pharma-Food Study on Extra Virgin Olive Oil Polyphenols. Foods 2022, 11, 1915. https://doi.org/10.3390/foods11131915

Thank you for the interesting manuscripts you mentioned. They are added in the manuscript.

Reviewer 4 Report

This manuscript is research on “Recovery of Antioxidant Compounds from Olive Pomace Using Green Extraction Processes”.  the research would be quite impressive, but unacceptable without major revision. comments are attached as a file

1.   

Minor editing of English language required

Author Response

Recovery of Bioactive Compounds from Olive Pomace Using Green Extraction Processes

Response to Reviewer 4

This manuscript is research on “Recovery of Antioxidant Compounds from Olive Pomace Using Green Extraction Processes”. the research would be quite impressive, but unacceptable without major revision. comments are attached as a file.

We would like to thank the Reviewer for the useful comments and suggestions made on our manuscript, which helped us improve it substantially. Please find below the changes made on our manuscript following the Reviewer’s recommendations.

  1. Please give an example of the analysis findings of the literature on classical extraction and soxhlet extraction methods in the introduction section as an antiradical effect and add a stronger related studies paragraph.

Conventional extraction is a widely used method for obtaining bioactive compounds from plants and other natural sources. It involves the use of solvents, such as ethanol or methanol, to extract the desired components from the plant material. This method is relatively simple and cost-effective, making it a popular choice in many research studies. On the other hand, Soxhlet extraction is a more specialized extraction technique that allows for the efficient extraction of target compounds from solid materials. It utilizes a continuous extraction process, where a solvent is repeatedly circulated through a sample. This method is particularly suitable for extracting compounds that have low solubility in the chosen solvent or for samples with complex matrices. Soxhlet extraction is known for its ability to yield high extraction efficiencies, making it advantageous when dealing with limited sample quantities. In summary, both conventional extraction and Soxhlet extraction methods have been widely used for the extraction of antioxidants with antiradical effects. These methods enable researchers to obtain natural extracts that possess significant antioxidant activity, providing valuable insights for the development of antioxidant-based therapies or dietary supplements to combat oxidative stress-related diseases.https://www.ncbi.nlm.nih.gov/pmc/articles/PMC7550548/  https://link.springer.com/article/10.1007/s11356-022-23337-6

  1. equation 1,2,3,4,5,6,7 please cite reference to each

Equations 1 and 2 does not need reference since are fundamental physicochemical equations. Equation 3 has been referred by Kyriakopoulou et al. (2013) as it is described in methodology. Equations 4-7 refers to Peleg model that is already referred in the relevant section.

  1. water and methanol mixture solvents, please specify v/v

We are not using a mix solvent system, each solvent was studied separately.

  1. Water or methanol was used when calculating the extraction yields. However, the results were not surprising. The water/methanol mixture is the commonly used solvent mixture in conventional extraction. At the same time, as the extraction time is optimum 60 minutes. Therefore, interpret the reasons for these two results in more detail and strengthen the interpretation contribution to the research.

Methanol and water were used separately as solvents in our extractions and water showed better results in the most of the cases. In addition, as shown by the extraction kinetics the best performance was indeed at 60min of extraction, but presented small differences with the lower extraction times especially of 20 min. After 20 min a plateau is reached in the majority of the cases. With the aim of energy consumption minimisation and the sustainability of our processes, lower extraction times are preferred since they exhibit approximately the same extraction yields as the higher extraction times.

  1. Please provide reference for the meaningful comment below. “The fact that extracts from untreated SOMW samples exhibited the highest yield for both water and methanol, indicates that moisture works collaboratively with the polar solvents enhancing the extraction of the desirable compounds. Moreover, high EYs were 0% 10% 20% 30% 40% 50% 5 10 20 60 EY (%d.b.) “

A reference is provided: DOI: 10.5772/intechopen.96092

  1. In Table 1, in section a, the efficiency of methanol in UEA compared to water solvent increased by almost 100%, but in MAE, it increased only from 2.45 to 2.88 in the antiradical effect of methanol and water solvent in MAE. Please interpret this difference in more detail by reference.

A paragraph explaining this trend is added: In general, ASD creates a rigid external layer in the dried material. Therefore, the penetration of solvent is more difficult (https://www.tandfonline.com/doi/abs/10.1080/10498850.2021.1900969 ). UAE through the cavitation phenomena can destroy this rigid layer permitting the effective penetration of the solvent  https://www.ncbi.nlm.nih.gov/pmc/articles/PMC7786612/ Therefore, in case of ASD the highest antioxidant capacity is observed in UAE extraction and especially when methanol is used as solvent, since methanol is more versatile and can extract a wider range of compounds, including both polar and nonpolar antioxidants.

  1. In the conclusion, please highlight the innovative propositions for this research more clearly.

A small paragraph indicating the innovation is added.

  1. Please add and comment on a table comparing the antiradical activity of the literature with this study. The following literature will help you Determination of theoretical calculations by DFT method and investigation of antioxidant, antimicrobial properties of olive leaf extracts from different regions G Baysal, EE Kasapbaşı, N Yavuz, Z Hür, K Genç, M Genç Journal of Food Science and Technology 58, 1909-1917

Thank you for this useful comment, your interesting recommendation and the manuscript you mentioned. However, the volume of the data in this study is already high and the authors believe that the addition of a table comparing the result of this study with others will increase the volume of this manuscript and may be confusing for the reader.

  1. the rate of plagiarism is quite high, please reduce it to an acceptable rate

We tried to reduce it, Please inform us if this is acceptable by you.

Round 2

Reviewer 1 Report

Thank you for accepting most of the comments. I still think the lack of the used phenols standards could be significant in explaining the chromatographic results. Maybe you could consider it in your future research?

Only one comment: in the present version, Line 622: try to find a somehow "lighter" word for "Estimated". Maybe "tentative" or similar? 

Reviewer 2 Report

No more comments

Reviewer 4 Report

I am pleased to inform accept of revision manuscript. 

Minor editing of English language required